# Transcriptional Regulation of zma-*MIR528a* by Action of Nitrate and Auxin in Maize

**DOI:** 10.3390/ijms232415718

**Published:** 2022-12-11

**Authors:** Eduardo Luján-Soto, Paola I. Aguirre de la Cruz, Vasti T. Juárez-González, José L. Reyes, María de la Paz Sanchez, Tzvetanka D. Dinkova

**Affiliations:** 1Departamento de Bioquímica, Facultad de Química, Universidad Nacional Autónoma de México, Ciudad de Méxcio 04510, Mexico; 2Department of Plant Biology, Swedish University of Agricultural Sciences, 75007 Uppsala, Sweden; 3Departamento de Biología Molecular de Plantas, Instituto de Biotecnología, Universidad Nacional Autónoma de Mexico, Av. Universidad 2001, Cuernavaca 62210, Mexico; 4Instituto de Ecología, Universidad Nacional Autónoma de México, Ciudad de México 04510, Mexico

**Keywords:** AuxRE, miR528, promoter analysis, TGA1, *Zea mays*

## Abstract

In recent years, miR528, a monocot-specific miRNA, has been assigned multifaceted roles during development and stress response in several plant species. However, the transcription regulation and the molecular mechanisms controlling *MIR528* expression in maize are still poorly explored. Here we analyzed the zma-*MIR528a* promoter region and found conserved transcription factor binding sites related to diverse signaling pathways, including the nitrate (TGA1/4) and auxin (AuxRE) response networks. Accumulation of both pre-miR528a and mature miR528 was up-regulated by exogenous nitrate and auxin treatments during imbibition, germination, and maize seedling establishment. Functional promoter analyses demonstrated that TGA1/4 and AuxRE sites are required for transcriptional induction by both stimuli. Overall, our findings of the nitrogen- and auxin-induced zma-*MIR528a* expression through *cis*-regulatory elements in its promoter contribute to the knowledge of miR528 regulome.

## 1. Introduction

Global analyses performed on several plant species have revealed that up to 90% of their genome is transcribed, although only a small fraction of the transcripts correspond to protein-coding RNAs (mRNAs) [1]. The remaining transcripts (non-coding RNAs, ncRNAs) have emerged as central regulators of different molecular programs implicated in plant development, growth, and stress response [2]. Several types of ncRNAs with different functions have been described, among which microRNAs (miRNAs) of 20 to 24 nt in length are known to control gene expression at the post-transcriptional level by inhibiting the accumulation or translation of specific mRNA targets [3]. Like protein-coding genes, the transcription of *MIR* genes is controlled by the action of the DNA-dependent RNA polymerase II (Pol II) along with general and specific transcription factors (TFs; extensively reviewed by [3]). A capped and polyadenylated primary miRNA transcript (pri-miRNA) is processed inside dicing bodies (D-bodies) formed by proteins such as DICER LIKE-1 (DCL1), double-stranded RNA-binding protein HYPONASTIC LEAVES (HYL1), and the structural protein SERRATE (SE), among others. Sequential processing originates the stem–loop precursor (pre-miRNA) that eventually produces a mature miRNA associated with the ARGONAUTE 1 (AGO1) protein to assemble the RNA-induced silencing complex (RISC; broadly reviewed in [3,4]).

Most plant miRNAs have been studied in their mature form with a primary focus on their target prediction and validation [5,6,7]. Their contribution to plant development and growth has been denoted through regulating particular targets that act as key players in controlling phase transition, biotic and abiotic stress response, nutrient homeostasis, and many other processes [8,9,10,11,12]. In contrast, just a few studies have focused on the regulation of *MIR* gene expression [13,14,15].

So far in maize (*Zea mays*), multiple miRNAs have been described as part of diverse cellular and developmental processes. One of them is zma-miR528, a monocot-specific miRNA, initially reported in rice. This miRNA displays a wide spectrum of functions by targeting distinct mRNA transcripts in different species [16,17,18,19,20,21]. Its function and transcriptional regulation have been more extensively studied in rice [15,17]. Os-miR528 is codified by a single gene and becomes highly accumulated upon vegetative to reproductive transition, opposite to the miR156 accumulation pattern [17]. In this work, the authors nicely demonstrated that Os-miR528 is regulated during plant development at the transcriptional level by the miR156 target Os-SPL9 and at the post-transcriptional level by aging-dependent alternative splicing of the pri-miRNA. They also found that *cis*-elements in miR528 gene promoters vary among different rice accessions, correlating with variable expression levels of the mature miRNA. Such complex and dynamic control of miR528 transcription awaits further exploration.

In maize, mature miR528 levels significantly increase during dry seed imbibition and drop upon germination completion [22]. It is also affected by external nitrogen supply, as seedlings grown under excess nitrogen conditions presented higher levels of this miRNA than those grown in a nitrogen-deficient media [19]. The regulation of specific miR528 targets, copper-containing laccase transcripts (*ZmLAC3* and *ZmLAC5*), affects lignin biosynthesis and correlates with plants being prone to lodging when miR528 accumulates due to high nitrate growing conditions. Furthermore, higher miR528 levels were observed during maize embryo development and maturation [23], as well as over the course of embryo dedifferentiation [24] during maize somatic embryogenesis (SE). Likewise, in well-established embryogenic calli, miR528 regulates the abundance of target mRNAs involved in oxidative stress, auxin accumulation, and differentiation. However, its level is reduced upon hormone depletion during in vitro plantlet regeneration [25].

Even though miR528 works as a central regulatory component in several pathways [26], little is known about the regulatory elements driving its expression in maize. Here we found that during seed germination of the maize cultivar Tuxpeño VS-535, miR528 is mostly derived from the expression of the zma-*MIR528a* locus. Exploration of the zma-*MIR528a* promoter region revealed the presence of conserved transcription factor binding sites (TFBS) related to diverse signaling pathways, including the nitrate (TGA1/4) and auxin (AuxRE) response networks. Accumulation of both pre-miR528a and mature miR528 was up-regulated by exogenous nitrate and auxin treatments during imbibition, germination, and seedling establishment. Partial 5′ promoter and specific TFBS deletion analyses demonstrated that TGA1/4 and AuxRE boxes are needed for transcriptional induction under both stimuli. Our findings contribute to understanding the regulatory network that controls zma-*MIR528* expression in response to previously reported stimuli and settles specific miRNA accumulation profiles in maize. This work also broadens the knowledge of factors regulating *MIR* gene expression in plants.

## 2. Results

### 2.1. Conservation of pre-miR528 in Monocots

To unveil the conservation among miR528 precursors in different species, we retrieved from PmiREN the sequences annotated as pre-miR528 in 21 monocot species, maize included (Appendix A). Multiple sequence alignment of precursors revealed a high degree of conservation, especially for the miRNA-5p, as all species presented the same sequence. On the contrary, the miRNA-3p region displayed more differences across the analyzed sequences. In addition, the alignment revealed a few conserved bases in the stem region (Appendix A). Therefore, we performed a phylogenetic analysis illustrating the evolutive relationships between precursors among species. Pre-miR528a and pre-miR528b from maize clustered in a well-defined clade together with species from the *Panicoideae* subfamily, like *Sorghum bicolor* (Sbi), *Panicum virgatum* (Pvi), *Setaria italic* (Sit), and *Saccharum* hybrid cultivar (Figure 1). Conversely, precursors, such as the one present in the banana (Mac-premiR528), showed a more distant evolutive relationship with sequences from maize, in agreement with previous reports [20]. The high degree of conservation is also noticeable by exploring the precursor folding, where sequences closely related to zma-premiR528a deployed mispairing at the 12th and 16th positions of the duplex region, leading to similar secondary structures (Appendix A). Altogether, these results indicate that precursors and mature miR528 are highly conserved among monocots at the sequence and structural levels, contrasting with their target diversity between species.

### 2.2. Identification of Promoter cis-Acting Elements in zma-MIR528a

Although multiple miRNAs have been described in maize, regulatory regions for most *MIR* genes remain elusive. According to miRBase (release 22.1) there are two different precursors for miR528 in maize, corresponding to zma-*MIR528a* and zma-*MIR528b* located on chromosomes 1 and 9, respectively [21]. Our first approach for their promoter description was to experimentally identify the transcription start site (TSS) by taking advantage of 5′-CAP modification in the primary transcript (pri-miR528a/b). Decapping and adapter ligation of imbibed seed RNA followed by RT and nested PCRs showed a fragment of about 300 nt, absent in the un-decapped sample (minus TAP control, Figure 2A). Cloning and sequencing of the amplification products allowed us to define the TSS position on the annotated sequence for *MIR528a*. The amplification product corresponding to pri-miR528b was not recovered, even though the 5′ RLM-RACE protocol was performed using primers matching both genes. Sequenced clones showed three alternative start sites with a preference for the adenine located 89 bp upstream of the pre-miR528a sequence previously annotated in databases (Figure 2B). To corroborate our findings, we analyzed public CAGE-Seq (Cap Analysis Gene Expression and deep Sequencing) data [27]. We found a sharp cluster of CAGE-TSS reads upstream of the pre-miR528a sequence, with the dominant TSS position concurrent with our 5′ RLM-RACE results (Appendix A).

Once the TSS was defined, we analyzed the promoter sequences of zma-*MIR528a* and homologous genes from different species to find putative TFBS (Appendix A). Only five of the analyzed sequences (zma-*MIR528a* included) presented a TATA box core element. In contrast, a higher number of genes displayed a CATT regulatory box within their promoter sequence (Appendix A). Furthermore, several *MIR528* promoter regions (~1500 bp upstream of TSS) displayed conserved binding motifs for TFs involved in cytokinin response (ARABIDOPSIS RESPONSE REGULATORS, ARR1), ABA signaling (ABA Response Elements, ABRE) [28], copper homeostasis (CuRE) [29], and vascular tissue formation and function (DNA-binding with one finger, Dof TFs) [30] (Figure 2C; Appendix A).

Notably, the zma-*MIR528a* promoter harbors two TGACG motifs known as binding sites for the TGACG-BINDING FACTORS 1/4 (TGA1/4), which are important regulatory components for nitrate responses in *A. thaliana* [31]. Also, the zma-*MIR528a* promoter presented one TGTCTC motif (Figure 2C), defined as an auxin response element (AuxRE), recognized by AUXIN RESPONSE FACTOR (ARF) proteins [32]. Both nitrate and auxin signaling routes are directly involved in reported experimental models where the accumulation of zma-miR528 is significantly enhanced [19,33]. Considering this, the TGA1/4 and AuxRE elements were more closely inspected in this study.

### 2.3. Accumulation of pre-miR528a Decreases as Seedling Establishment Takes Place

To address how zma-*MIR528a* is transcriptionally regulated, the presence of precursor transcripts should be evaluated. Previous reports indicate that mature miR528 highly accumulates throughout maize embryonic development, with further increase upon seed imbibition and decline at post-germinative stages [21,22]. Thus, we explored whether miR528 precursors could be detected during germination and seedling establishment until 72 h after seed imbibition (Figure 3A). Although precursors from both zma-*MIR528a* and zma-*MIR528b* were evaluated (positive amplification from genomic DNA; gDNA), only pre-miR528a (123 nt) was detected by end-point RT-PCR (Figure 3B). Relative quantification by qRT-PCR showed that pre-miR528a abundance continuously drops from dry seed to initial seedling stages (Figure 3C). By contrast, mature miR528 accumulated towards the first 24 h of imbibition and then went down (Figure 3D), consistent with the aforementioned reports. Overall, these observations allowed us to select germination and seedling establishment as our working model to evaluate the effect of diverse stimuli upon zma-*MIR528a* expression by quantifying its precursor.

### 2.4. Exogenous Nitrate and Auxin Treatments during Maize Seed Imbibition Trigger Increases in pre-miR528a and Mature miR528 Levels

After establishing the experimental model, we tested two different exogenous treatments during germination and early seedling growth to evaluate their effect on the precursor and mature miR528 accumulation (Figure 4A). Incubation with 30 mM KNO_3_ positively impacted the seed germination rate (Figure 4B top and Appendix A). In agreement with previous reports [34], extended incubation with nitrate accelerated seedling growth and shoot expansion (Appendix A). On the other hand, NPA-treated seedlings subjected to a short pulse (6 h) of the auxin NAA did not show evident growth differences with respect to its control (NPA alone) and contrasted with H_2_O imbibition (Figure 4B bottom and Appendix A). However, extended incubation with NAA promoted the emergence of multiple lateral roots (Appendix A), confirming the reported physiological effect of exogenous auxin [35].

As expected according to previous studies [19], significant increments of pre-miR528a and miR528 were observed in the presence of nitrate solution relative to H_2_O at 24, 48, and 72 h upon seed imbibition (Figure 4C). The treatment with NAA also increased their levels, with the highest accumulation observed at 4 and 6 h after the auxin pulse on seedlings pre-treated with NPA. On the other hand, inhibition of polar auxin transport by NPA alone did not cause significant changes on either precursor or mature miRNA levels, since at 72 h of imbibition they showed considerably reduced levels (Figure 3C,D and Figure 4D). Owing to this, we tested if the observed increments were due to transcriptional events. As shown in Appendix A, nitrate and auxin treatments involved transcriptional activity to promote the highest accumulation of pre-miR528a and mature miR528. The inclusion of alpha-Amanitin (inhibitor of RNA pol II, a-aman) exhibited a significant reduction in the levels of both molecules with respect to the treatment alone. Subsequently, we tested whether up-regulation of zma-*MIR528a* caused by nitrate and auxin altered the accumulation of previously confirmed mRNA targets [21]. The abundance of *BASIC HELIX-LOOP-HELIX 152* (*bHLH152*; Zm00001d016873), *MULTIDRUG AND TOXIC COMPOUND EXTRUSION/BIG EMBRYO 1* (*MATE/BIGE1*; Zm00001d012883), and *SUPEROXIDE DISMUTASE 1a* (*SOD1a*; Zm00001d031908) transcripts significantly decreased in samples treated either with KNO_3_ or NAA (Appendix A). Overall, these results show that nitrate and auxin trigger transcription of zma-*MIR528a*, resulting in significant precursor and mature miRNA accumulation as well as target down-regulation during maize seed imbibition.

### 2.5. zma-MIR528a Promoter Harbors Regulatory Elements Responsible for Transcription Enhancement in Response to Nitrate and Auxins

Given that nitrate and auxin treatments triggered zma-*MIR528a* expression, we sought to analyze the regions within the promoter responsible for such induction. From our previous TFBS analysis, the presence of TGA1/4, AuxRE, and TATA boxes was considered to design nested 5′ deletion segments of the promoter fused to the eGFP/GUS reporters (Figure 5A). The resulting constructs were used to perform transient activity assays in maize protoplasts and showed activity for both reporters under the full-length promoter (pMIR_1, −1180 bp from TSS; Appendix A). This indicated that the zma-*MIR528a* promoter works as an integral and functional sequence. However, as the sequence was narrowed, transcriptional activity decayed. Deletion of 814 nt (pMIR_3) and 1157 nt (pMIR_4) upstream of the TSS significantly reduced the promoter activity under control conditions (Appendix A). Next, we assayed the activity of each construct when transfected protoplasts were incubated with nitrate (10 mM KNO_3_) and auxin (1 μM NAA) treatments. An increase in GUS reporter expression under nitrate stimulus was observed for protoplasts transfected with the full-length (pMIR_1) and narrowed p_MIR2 (-843 bp from TSS) promoter versions (Figure 5B). In contrast, only the full-length promoter (pMIR_1) presented significant induction with NAA (Figure 5C). When nitrate and auxin were used in a combined treatment, the pMIR_1 and pMIR_2 constructs also significantly increased their transcriptional activity (Figure 5D). Since the pMIR_2 construct lacks one TGA1/4 box and the AuxRE but still harbors the proximal TGA1/4 box, this result suggests that this motif is sufficient to induce expression at least in response to high nitrate concentration.

### 2.6. TFBS within the zma-MIR528a Promoter Contribute Differentially to Nitrate and Auxin Induction

Although transient expression assays evidenced that the zma-*MIR528a* promoter harbors *cis*-acting sequences involved in transcriptional activation in response to nitrate and auxin treatments, the contribution of each TGA1/4 box and the AuxRE element was not completely clear due to the likely presence of additional response elements within the promoter. Therefore, we tested the effect of individual TFBS deletions in the context of a full-length promoter. In contrast to sequential deletions, single TFBS deletions did not affect basal transcription of the reporter (mock in Figure 6), supporting a role of other elements within the region deleted in pMIR_2 that guide expression under control conditions.

Interestingly, transcriptional induction under WT promoter was significantly greater for combined nitrate and auxin, as compared to single treatments (Figure 6). Deleting the distal TGA1/4 site (−1065 position; ∆TGA1) still showed significant induction by nitrate, auxin, or combined treatments. Conversely, deletion of the TGA1/4 site at position −738 (∆TGA2) abolished the induction by either of the individual treatments but still allowed significant, although minor, increase by the combined action of nitrate and auxin (Figure 6). Altogether, these results indicate that the TGA1/4 site proximal to the TSS plays a pivotal role in zma-*MIR528a* transcriptional activation by either nitrate or auxin. Finally, the mutation of the TGTCTC sequence (∆AuxRE, position −875) negatively affected only the induction by NAA alone, proving it is required for proper activation of the zma-*MIR528a* promoter by the auxin signaling pathway (Figure 6).

### 2.7. ARF34 Contributes to Transcriptional Activation of the zma-MIR528a Promoter

To further explore the role of auxin response machinery on zma-*MIR528a* promoter activation, we co-transfected maize protoplasts with pMIR_1 or mutant reporter constructs along with an effector plasmid containing the ARF34 coding sequence (Zm00001eb031700, Figure 7A), which is a well-studied transcriptional activator ARF [36]. Significantly higher induction was evident for protoplasts co-transfected with this effector and pMIR_1 reporter (Figure 7B and Appendix A). An increase in reporter activity was not detected for protoplasts co-transfected with ARF4 (Zm00001eb067270) activator [36], suggesting that the zma-*MIR528a* promoter could be particularly responsive to ARF34. Furthermore, induction was observed for co-transfections with ∆TGA1 or ∆TGA2 constructs, but not with the construct without the AuxRE site (∆AuxRE + ARF34; Figure 7B). Interestingly, the induction was significantly lower for ∆TGA2 than the one observed for pMIR_1 or ∆TGA1 reporters. This suggests that the proximal TGA1/4 and AuxRE motifs might cooperate for full auxin inducibility of the zma-*MIR528a* promoter. Finally, the presence of a 1 µM NAA further increased the transcriptional activity of the wild-type promoter (pMIR_1) in the presence of ARF34 (Figure 7C), supporting the involvement of additional auxin signaling pathway components in zma-*MIR528a* regulation. Albeit our results exposed the relevance of AuxRE and proximal TGA1/4 sites for NAA-mediated activation of the zma-*MIR528a* promoter, direct interaction between these elements and ARF34 or other transcription factors will need further experimental demonstration.

## 3. Discussion

Although several miRNAs have been studied in plants, there is a lack of information about promoter regions and regulatory elements that coordinate their expression. Their analysis will help to better understand their regulation and how different stimuli control their expression. We experimentally validated the transcription start site (TSS) for the zma-*MIR528a* gene at 89 bp upstream from the mapping site of the precursor pre-miR528a. Comparable lengths have been reported for *A. thaliana* and rice *MIR* genes, as most miRNAs display less than 200 nt between the TSS and the stem–loop-structured precursor [37]. We found additional start sites adjacent to the predominant TSS, consistent with next-generation data previously reported [27]. Narrow clustering of multiple TSS is a common feature among genes with tissue-specific expression [27], which could explain the specific accumulation patterns reported for miR528 in the vascular tissue of maize leaves, stems, and immature embryos [19,21,38]. The TSS determination enabled us to limit and more accurately scrutinize the promoter region for putative *cis*-regulatory elements (CREs). Interestingly, both zma-*MIR528a* and zma-*MIR528b* displayed core promoter regulatory elements analogous to those present in other *MIR* genes found in rice [39] and *A*. *thaliana* [40]. However, only zma-*MIR528a* presented a TATA box element nearby the TSS region. This element has been reported to affect the promoter strength and increase the expression of endogenous and synthetic genes in plants [41] and is over-represented in *MIR* plant promoters [42]. Therefore, we hypothesized that TATA box presence in *MIR528a* could be involved with the differential expression rates between both *MIR528* genes in maize, explaining our lack of detection for pre-miR528b. Also, expression of *MIR528b* may be restricted to particular tissues or developmental stages, even though diverse expression data sources have only collected reads mapping against the *MIR528a* genomic region [43,44,45]. Moreover, numerous CAAT sites were found close to the TSS and throughout the promoter for both genes in maize. CAAT boxes are conserved sites believed to determine the efficiency of transcription [46]. Indeed, the insertion of multiple CAAT boxes is a proposed model for acquisition of promoter activity for some *MIR* genes such as *MIR1444*, *MIR058*, and *MIR12112* in *Vitis vinifera* [47]. In addition, these core promoter elements were conserved in homologous genes, which suggests they may have a general role in guiding the expression of *MIR528* in monocots.

A higher degree of conservation was found among pre-miRNAs at the sequence and structural levels, as each precursor leads to identical mature miRNA in the analyzed sequences. This shared secondary structure might influence the processing and the accumulation of miR528 among species, as precise cleavage of pre-miRNA requires structural and sequence determinants, but their relative contributions are still being studied [48,49].

The presence of specific TFBS within the promoter provided hints about the metabolic and physiological pathways commanding zma-*MIR528a* expression. We found two TGA1/4 (also known as *activation sequence-1*, AS-1) binding motifs in the zma-*MIR528a* promoter that were highly conserved in other *MIR528* genes. The TGA1/4 transcription factors are members of the basic leucine zipper (bZIP) family and participate in complex transcriptional networks associated with nitrate response and salicylic acid biosynthesis [31,50,51]. zma-*MIR528a* also exhibited an auxin response element (AuxRE), even though its presence was less conserved among homologs. The auxin responsiveness is achieved by the interaction of ARFs with AuxREs at promoter regions of diverse genes involved in several developmental processes [32,52,53]. TGA1/4 and AuxRE sites are found in the promoters of several *Zea mays* sequenced accessions (Appendix A) supporting their relevance for zma-*MIR528a* expression regulation in response to external stimuli. In addition, multiple TFs that potentially recognize TGA1/4 and AuxRE sites exhibited expression profiles similar to *MIR528a* expression in distinct tissues (Appendix A). Accordingly, we found that high nitrate conditions and exogenous auxin application increased the zma-*MIR528a* precursor and mature miRNA levels at early seedling stages of Tuxpeño VS-535 maize. Previous reports indicate that mature miR528 levels are affected in maize roots under nitrate starvation [19,38], while nitrogen luxury enhanced its expression [19] in different maize lines. High nitrate and auxin concentrations are commonly used for callus induction and proliferation during maize somatic embryogenesis (SE), which might explain the high miR528 abundance in dedifferentiated tissues [21,25,33].

The responsiveness of miRNAs to nitrate and auxin is well-documented in diverse plants and experimental models [38,54]. For instance, treatments with indoleacetic acid (IAA) promote transcription of ath-*MIR393b* in aerial seedling organs [55]. Furthermore, miR393 also accumulates two hours after incubation in 5 mM potassium nitrate solution and induces target down-regulation [56,57]. Interestingly, during SE induction in *A. thaliana,* expression of *MIR393a* and *MIR393b* increase to enable explant sensitivity to auxins [58], which mirrors the miR528 response in maize SE.

Transient expression assays with the zma-*MIR528a* promoter confirmed that elements responsive to nitrate and auxins are localized within the analyzed sequence. The distribution of TGA1/4 sites in zma-*MIR528a* is similar to footprints detected in promoters of genes transcriptionally activated by nitrate in Arabidopsis [59]. However, only the removal of the TGA1/4 site nearest to the TSS (ΔTGA2) significantly affected the transcription enhancement by external stimuli, evidencing a differential contribution of each site. This is congruent with studies reporting that higher gene expression correlates with TGA1/4 binding motifs located closer to TSS [59,60]. In addition, our results suggest that the proximal TGA1/4 site in the zma-*MIR528a* promoter also cooperates with auxin induction. Enhancement of activity by NAA was not observed for the ΔTGA2 construct, in spite of the presence of AuxRE. This is not surprising, since several TFs that bind the TGACG motif guide gene expression in response to synthetic auxins, methyl jasmonate, and H_2_O_2_, acting as molecular linkers between different signaling pathways [61,62,63,64].

Early auxin response genes tend to have two or more adjacent TGTCTC/TGTCGG motifs separated by a variable spacer sequence to allow ARFs’ dimerization and proper gene activation [32,65,66]. Nonetheless, nearly 86% of the ARFs’ targeted sites in maize are composed of two or more TGTC core motifs with less than 50 nucleotides of spacing [36,67]. Interestingly, the zma-*MIR528a* promoter displays a TGTC core motif located 36 nt downstream of the analyzed AuxRE site, presenting architectural features analogous to other auxin-induced genes in maize [36]. Transcriptional activator ARF34 modulates the activation of the zma-*MIR528a* promoter through the AuxRE motif, as shown by our transactivation assays in protoplasts. Hence, alteration of the architecture comprising AuxRE and TGTC sites might abolish auxin signaling over zma-*MIR528a*. Similar results were reported for Ath-*MIR390a*, as deletion of a 36 bp portion containing an inverted AuxRE motif suppressed the response of this gene to auxin treatments [14].

Overall, the analysis of the zma-*MIR528a* promoter suggests that nitrate and auxin pathways cooperate to induce *MIR528a* expression (Figure 8). This synergism may occur due to an overlap of the molecular regulatory routes previously reported for NO_3_^−^ and auxins. The primary nitrate response proceeds through master regulators (TCP-NLP) that directly control TFs such as TGA1/4 to localize their binding sites in promoters of target genes. They are also implicated in auxin homeostasis and signaling by regulating *TRYPTOPHAN AMINOTRANSFERASE OF ARABIDOPSIS 1* (*TAA1*), involved in IAA biosynthesis [68], and promoting the accumulation of AUXIN SIGNALING F-BOX 3/TRANSPORT INHIBITOR RESPONSE 1 (AFB3/TIR1), required for degradation of ARFs’ repressor Aux/IAA [68,69]. Considering this, the co-existence of high auxin and nitrate conditions enhances zma-*MIR528a* expression through localizing TGA1/4 and ARF TFs at cognate motifs in the promoter region, which results in the reduction of mRNA targets levels (Figure 8). Such outcomes have been reported for the stimuli studied in this work, resulting in maize plant lodging under high nitrate growth conditions [19] or callus induction and proliferation from immature embryos [21,25]. Moreover, auxins accumulate during late embryo development and regulate seed dormancy acquisition in several plants [70,71]. This auxin increment might be guiding the transcriptional activation of zma-*MIR528a*, as the precursor is highly accumulated in dry maize seeds. However, additional regulatory routes, such as the ROS or the cold stress response pathways could also control zma-*MIR528a* expression by activating other *cis*-regulatory elements. Furthermore, the regulatory mechanism might depend on the species, developmental stage, or condition being studied [15,17].

Transcriptional activation of zma-*MIR528a* during late embryogenesis correlates with high pre-miR528 levels in dry seeds and mature miR528 transient accumulation upon seed imbibition, followed by rapid decrease after germination. This is accompanied by fine target regulation during the process [21], which possibly contributes to appropriate developmental and stress responses in early germination [72]. However, zma-*MIR528a* knock-out maize mutants are required to confirm its physiological role in these processes. Nevertheless, this work provides critical information on the transcription of a developmentally relevant microRNA and identifies the key TF binding sites that contribute to the auxin and nitrogen response of *MIR528a*. Future work should also explore additional layers of regulation for *MIR528* expression to increase knowledge about the relationship between factors determining miR528 accumulation and target regulation, as well as the implication of complex regulatory networks on the growth and development of different monocot plants.

## 4. Materials and Methods

### 4.1. Plant Material and Growth Conditions

The Mexican maize (*Zea mays* L.) cultivar Tuxpeño VS-535 was used due to its agronomic relevance and the available previous characterization of miR528 in SE [21]. Seeds were disinfected with 6% commercial NaClO solution under constant agitation (250 rpm) for about 5 min and rinsed five times with 100 mL of fresh sterilized water. Imbibition, vertical germination, and seedling establishment were performed as previously reported [35]. Briefly, seeds were placed in paper towel rolls and incubated vertically in cylindrical containers with a 12 h light/12 h dark photoperiod at 28 °C. Nitrate stimulus application was conducted following previous reports [19] with some modifications. Rolls were soaked in 150 mL of either sterile water (control condition; mock) or 30 mM KNO_3_ solution for 72 h, with sampling after 24, 48, and 72 h of continuous treatment. For exogenous auxin addition, experiments were performed as recommended by [35]. Seeds were imbibed for 72 h in water (mock) or 50 µM N-1-naphthylphthalamic acid (NPA, auxin transport inhibitor). After this time, a portion of the NPA-treated seedlings was transferred to 50 µM 1-Naphthalene Acetic Acid (NAA), and samples were collected at 0, 2, 4, and 6 h after the NAA pulse. Embryo axes and coleoptiles were extracted from seeds and seedlings and immediately frozen in liquid nitrogen until further use. Extended treatment with 50 µM NAA for 48 h was performed for phenotype registration. Controls remained under H_2_O or 50 µM NPA alone for each time point. All reagents were purchased from Merck, Sigma-Aldrich, St. Louis, MO, unless otherwise stated.

### 4.2. RNA Isolation, Size Fractionation, and Purification

Total RNA was extracted from collected tissues using TRIzol reagent (Invitrogen, Thermo Fisher Scientific Inc. Waltham, MA, USA) as previously reported [73]. DNase I treatment, size fractionation, and concentration of large (>200 nt) and small (<200 nt) RNAs were performed using the RNA Clean and Concentrator kit (Zymo Research, Irvine, CA, USA) following the manufacturer’s instructions.

### 4.3. Precursor, Mature microRNA, and Target Quantification by RT-qPCR

For pre-miRNA detection, 2 µg of the fraction of small RNAs (<200 nt) were polyadenylated using the Poly(A) tailing kit (Invitrogen, Waltham, MA, USA). Reverse transcription (RT) reaction was carried out with oligo (dT) and the Improm-II reverse transcription system (Promega, Madison, WI, USA). Primers for each miR528 precursor were designed using Primer3Plus [74], with qPCR settings activated. Both precursors (pre-miR528a/b) were accessible for amplification using a combination of premiR_Fw and preMIR528_Rv3 oligonucleotides, as they were designed to align to conserved sites presented in both sequences (Appendix A). For the mature miRNA, previously reported stem–loop and specific forward primers were used along with pulsed stem–loop RT reactions for miR528 and U6 snRNA using the sRNA fractions [21]. For target quantification, total RNA was reverse transcribed (RT) using oligo (dT) and the Improm-II reverse transcription system (Promega, Madison, WI, USA). For each target, previously reported primers were used [21]. qPCR was performed using the Maxima SYBR Green/ROX qPCR Master Mix in a 7500 Real-time PCR System (Applied Biosystems, Bedford, MA, USA). Relative expression was calculated by the 2^−∆∆Ct^ method with normalization to 18S rRNA (precursor and targets) or U6 snRNA (for miRNA) as internal housekeeping gene controls. Data from each experiment (three independent experiments with three technical replicates each) were subjected to one-way analysis of variance (one-way ANOVA) with Tukey’s Multiple Comparison post hoc test for statistical significance.

### 4.4. Experimental Validation of the Transcription Start Site (TSS) by 5′ RLM-RACE

Mapping of TSS was performed following a reported protocol [75] with slight modifications. About 10 µg of total RNA was treated with the QuickCIP enzyme (NEB, Ipswich, MA, USA) at 37 °C for 1 h. Then, 15 µL of 5 M ammonium acetate was added to stop each reaction; treated RNA was purified by phenol-chloroform extraction and isopropanol precipitation. Decapping enzyme (NEB, Ipswich, MA) was used for 5′-CAP removal of the treated RNAs. Minus TAP reactions were also assembled as ligation controls. Ligation of 5′ RACE adapter (Appendix A) was performed employing 2 µL of CIP-treated and decapped RNA and 2 µL of T4 RNA ligase (2.5 U/µL) for 1 h at 37 °C. The ligation products were reverse transcribed using oligo (dT) and the Improm-II reverse transcription system (Promega, Madison, WI, USA). Consecutive amplification reactions (nested PCR) were conducted using adapter-specific and gene-specific primers (Appendix A). The final PCR products were separated by agarose gel electrophoresis and purified using the Wizard^®^ SV Gel and PCR Clean-Up System (Promega, Madison, WI, USA). Finally, the purified product was cloned into the pGEM-T easy vector (Promega Madison, WI, USA), and the resulting clones were sequenced.

### 4.5. Phylogenetic Analysis of miR528 Precursors and Promoters in Monocots

Precursor sequences of miR528 were obtained for 21 monocot species (Appendix A) using miRBase release 22.1 [76] and the Plant microRNA Encyclopedia PmiREN version 2.0 [77]. Upstream regions (≈2000 nt from the precursor mapping site) were also retrieved from the Ensembl Plant database release 52 [78] for promoter homology comparisons. Sequences were aligned using the nucleotide-optimized MUSCLE algorithm [79] with 50 iteration cycles. The phylogenetic reconstructions were conducted using the MEGA X [80] and Maximum Likelihood (ML) method with the best-adjusted model (*3-parameter Tamura*) and 1000 bootstrap replications. In addition, folding of some precursor sequences was predicted using the webserver RNAFold from the University of Vienna (http://rna.tbi.univie.ac.at/cgi-bin/RNAWebSuite/RNAfold.cgi, accessed on 9 May 2022).

### 4.6. Promoter cis-Element Analysis

In silico analysis of cis-regulatory elements in the upstream region of *MIR528* promoters was performed with the NEW PLACE (https://www.dna.affrc.go.jp/PLACE/?action=newplace, accessed on 18 March 2022) online analysis software PlantPAN3.0 (http://plantpan.itps.ncku.edu.tw, accessed on 18 March 2022) [81] and PlantCARE (http://bioinformatics.psb.ugent.be/webtools/plantcare/html/, accessed on 18 March 2022) [82].

### 4.7. Generation of Plasmid Constructions

Genomic DNA was extracted from maize seedlings using a reported CTAB method [83]. Specific primers for the full-length *MIR528a* promoter (pMIR_1; Appendix A) were used to amplify the region between the position −1510 upstream of TSS and the precursor sequence (209 nt beyond TSS). Amplification was accomplished using the Phusion High-Fidelity DNA Polymerase (NEB, Ipswich, MA, USA) following the manufacturer’s protocol. The amplicon was gel-purified using the Wizard^®^ SV Gel and PCR Clean-Up System (Promega, Madison, WI, USA) kit and cloned into the pGEM-T easy vector (Promega, Madison, WI, USA) for sequencing.

Sequential deletion fragments of the promoter were amplified with specific primers (pMIR_1-4; Appendix A) using the pMIR_1 plasmid as template and were cloned into the pENTR-D-TOPO vector (ThermoFisher, Waltham, MA, USA). Proper orientation was verified by PCR. Correct entry clones were used for sub-cloning into the pBGWFS7.0 vector employing the Gateway LR Clonase II Enzyme Mix (Invitrogen, Waltham, MA, USA). The integrity of constructs and reporter genes was assayed by sequencing. Deletion of binding sites (TGA1/4 and AuxRE) was achieved by site-directed mutagenesis using the overlapping PCR method, as reported by [84]. WT and mutant expression cassettes were cloned into the pGEM-T easy vector for Sanger sequencing and subsequent promoter activity assays.

Effector plasmids for co-transfection assays were obtained by Gateway recombination of the entry clones pUT6075 and pUT3104 (acquired from the ABRC) corresponding to ZmARF34 (Zm00001eb031700) and ZmARF4 (Zm00001eb067270), respectively, with pEarleyGate 102 as the destination vector using the LR-clonase enzyme mix (Invitrogen, Waltham, MA, USA). Correct insertion and integrity of the corresponding ARF coding sequences downstream of the CaMV 35S promoter were checked by Sanger sequencing.

### 4.8. Maize Leaf Protoplast Isolation and Transfection

Protoplast isolation was carried out following the protocol reported by [85] with modifications. Briefly, the fully expanded second leaves from etiolated maize plants were cut into ≈ 0.5 cm strips. Leaf strips were soaked, vacuum infiltrated with the enzyme solution (1.5% cellulase R10, 0.75% macerozyme R10 (Yakult, Tokyo, Japan), 0.6M mannitol, 10 mM MES (pH 5.7), 10 mM CaCl_2_, and 0.1% BSA), and incubated for 3 h with constant agitation (50 rpm) in darkness. Protoplasts were collected by filtration and centrifugation (200× *g* for 4 min), washed out twice with W5 solution (154 mM NaCl, 125 mM CaCl_2_, KCl 5 mM, and 2 mM MES pH5.7), and resuspended into cold MMg solution (15 mM MgCl_2_, 0.4 M mannitol, 4 mM MES, pH 5.7) to a final concentration of 10^6^ mL^−1^ protoplasts. For transfection, about 2 × 10^5^ protoplasts were combined with 50 µg of each plasmid DNA and PEG transfection solution (40% *w*/*v* PEG 4000, 0.6 M mannitol, and 0.1 M CaCl_2_) and incubated for 30 min at 25 °C covered from light. Next, the transfected cells were washed with W5 buffer and incubated at 25 °C in the dark for 16 h for basal expression detection. To avoid protoplast swelling and bursting, lower concentrations of each treatment (10 mM KNO_3_, 1 µM NAA, or both) were applied for the last 6 h of incubation as reported by [86] and [87]. After this, the protoplasts were harvested for further analysis. Transfection efficiency and normalization were assessed by absolute qPCR based on the number of copies of recombinant DNA successfully delivered into protoplasts after transfection, as previously reported [88].

### 4.9. Promoter Activity Analysis

Promoter activity was determined for each construct by measuring GUS enzymatic activity with 4-Methylumbelliferyl-β-D-glucuronide (4-MUG) as substrate. After incubation and treatments, protoplasts were harvested, and protein extracts were collected following the protocol reported by [89]. Fluorescence was measured based on a 4-methylumbelliferone (4-MU) standard curve using the Varioskan LUX Multimode microplate reader (ThermoFisher, Waltham, MA, USA). In each sample, the total protein concentration was estimated by Bradford assay. GUS activity was calculated based on the 4-MU standard curve and expressed in nmol 4-MU min^−1^ mg^−1^.

## Figures and Tables

**Figure 1 ijms-23-15718-f001:**
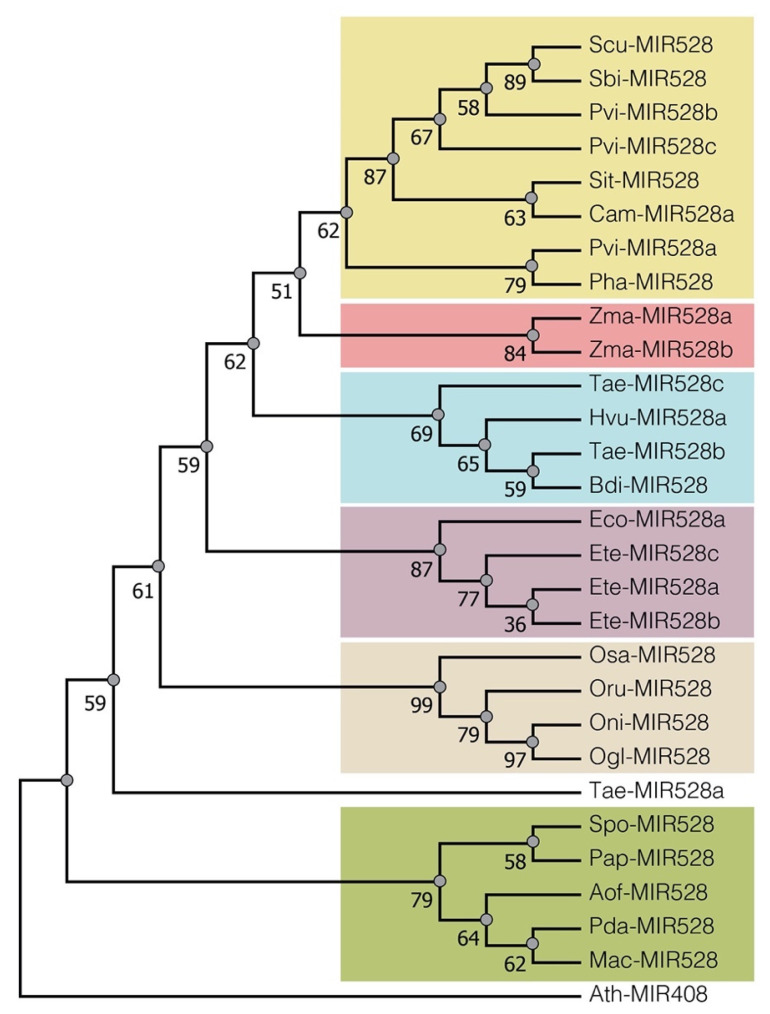
Phylogenetic analysis of pre-miR528 sequences found in monocots. Phylogenetic reconstruction was made with MEGA X using the Maximum Likelihood (ML) method and the best-adjusted model (*3-parameter Tamura*) with 1000 bootstrap iterations. Twenty-eight monocot miR528 precursor sequences were clustered in different subclades using the ggtree package in R studio. *Arabidopsis thaliana* miR408 precursor was used as an outlier with an equivalent function, as this miRNA shared several targets with miR528 [20]. The bootstrap values are marked as numbers at nodes. Aof: *Asparagus officinalis*, Bdi: *Brachypodium distachyon*, Cam: *Cenchrus americanus*, Eco: *Eleusine coracana*, Ete: *Eragrostis tef*, Hvu: *Hordeum vulgare*, Mac: *Musa acuminata*, Ogl: *Oryza glaberrima*, Oni: *O. nivara*, Oru: *O. rufipogon*, Osa: *O. sativa*, Pha: *Panicum hallii*, Pvi: *P. virgatum*, Pap: *Phalaenopsis Aphrodite*, Pda: *Phoenix dactylifera*, Scu: *Saccharum* hybrid cultivar, Sit: *Setaria italica*, Sbi: *Sorghum bicolor*, Spo: *Spirodela polyrhiza*, Tae: *Triticum aestivum*, Zma: *Zea mays*.

**Figure 2 ijms-23-15718-f002:**
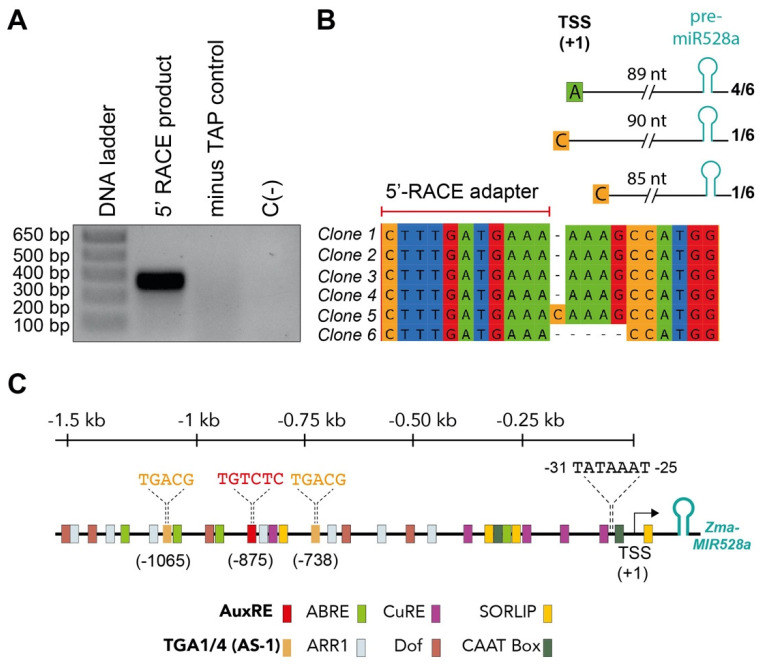
Identification of *cis*-acting elements within the zma-*MIR528a* promoter. (**A**) The *MIR528a* TSS was mapped by 5′RLM-RACE as described in methods. Nested PCR products separated by gel electrophoresis are shown. (**B**) Sequencing results of *MIR528a* 5′-RACE clones with identified TSS (+1) are shown in the upper scheme. The distance between TSS and pre-miR528a is indicated; the number of clones identified with the same 5′ end, as a proportion of the total sequenced clones, is shown at the right. (**C**) The 1500 bp of genomic DNA sequence upstream of zma-*MIR528a* TSS was analyzed using PlantCare and New PLACE databases. Different color markers indicate major predicted *cis*-acting elements. The sequence and position of two TGA1/4 and one AuxRE elements are highlighted.

**Figure 3 ijms-23-15718-f003:**
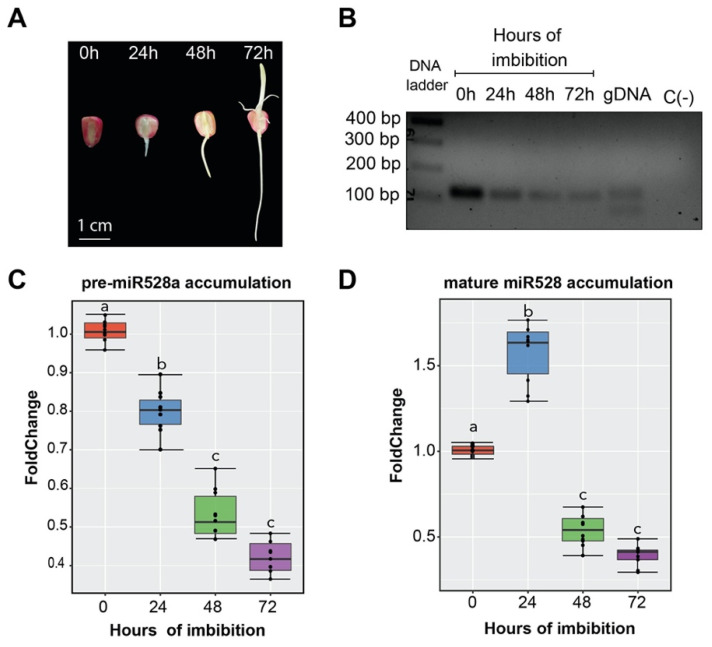
Accumulation levels of zma-*MIR528a* precursor upon seed germination. (**A**) Maize VS-535 seeds were germinated in vertical chambers, and seedling emergence was registered upon 0, 24, 48, and 72 h after imbibition. (**B**) Detection of pre-miR528a by end-point RT-PCR in samples collected at each timepoint. Oligonucleotides were designed to amplify both pre-miR528a (123 nt) and pre-miR528b (78 nt), as shown for genomic DNA (gDNA). (**C**,**D**) Quantification by real-time PCR of pre-miR528a and mature miR528 in maize embryonic axes at the indicated stages of seed imbibition. Fold change represents the abundance relative to dry seed (0 h) normalized either by 18S rRNA (precursor) or U6 snRNA internal control (mature miRNA). Error bars represent the standard error of the mean from three biological replicates with three technical replicates for each one (*n* = 9). Data were analyzed by one-way ANOVA with multiple comparisons by the Tukey post hoc test. Boxes that do not share at least one identical letter differ significantly (*p* < 0.05) from each other.

**Figure 4 ijms-23-15718-f004:**
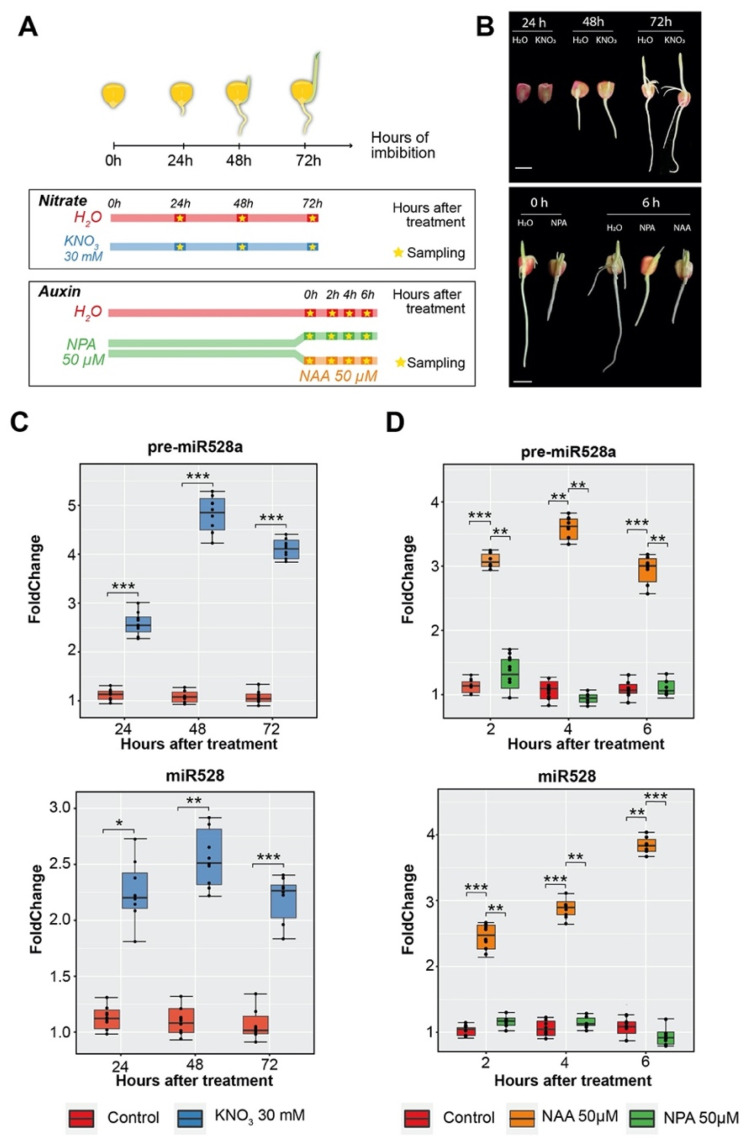
The effect of exogenous nitrate and auxin treatments on pre-miR528a and mature miR528 accumulation during maize seed imbibition. (**A**) Experimental design for treatment application during seed imbibition and seedling establishment. To evaluate high nitrate concentration, maize seeds were incubated in water (control) or 30 mM KNO_3_ for 72 h with sampling at 24, 48, and 72 h of continuous treatment. For exogenous auxin application, seeds were imbibed for 72 h in water (control) or 50 µM N-1-naphthylphthalamic acid (NPA, auxin transport inhibitor). Afterward, a portion of the NPA-treated seedlings was transferred to a 50 µM 1-Naphthalene Acetic Acid (NAA) solution and samples were collected at 0, 2, 4, and 6 h after the auxin pulse. (**B**) Representative images for maize seed germination and seedling establishment at 24 h, 48 h, and 72 h of imbibition under high nitrate condition (top) or 72 h imbibed seedlings at 6 h after the application of exogenous auxin (bottom). Scale bar = 1 cm. (**C**) Relative accumulation levels of miR528 precursor and mature form in samples treated with nitrate. (**D**) Relative accumulation levels of miR528 precursor and mature form in samples treated with exogenous auxin. The expression levels were normalized using 18S rRNA (for precursor) and U6 snRNA (for mature miRNA) accumulation. Error bars represent standard errors of means from at least three independent replicate experiments. * *p* < 0.05, ** *p* > 0.01, and *** *p <* 0.001 (Data were analyzed by one-way ANOVA with Tukey post hoc test comparing treated samples to control for each time point).

**Figure 5 ijms-23-15718-f005:**
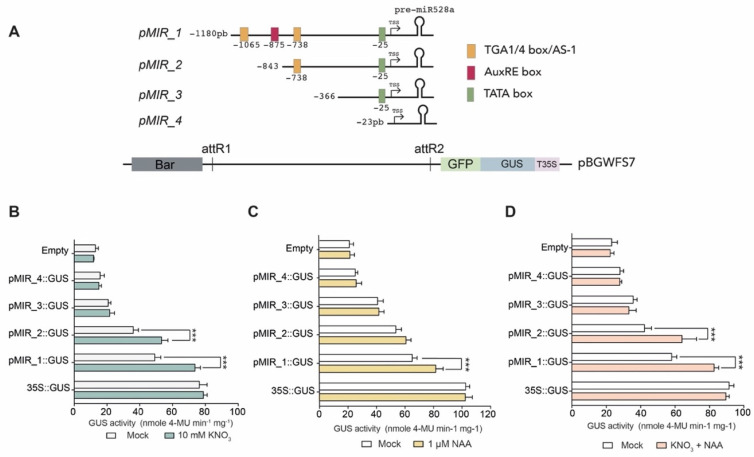
Dissection of *cis*-elements required for transcriptional up-regulation of miR528 by nitrate and auxins. (**A**) Constructs were generated including the *MIR528a* full promoter (pMIR_1) or 5′ consecutive deletions (pMIR_2-4) upstream of eGFP and GUS reporter genes (green and blue boxes, respectively). The nucleotide position of each deletion fragment is indicated at the left on each panel. Colored boxes represent relevant *cis*-acting elements. (**B**–**D**) GUS activity was evaluated by fluorometric assay on maize protoplasts transfected with each construct under nitrate (**B**), auxin (**C**), or a combination of both treatments (**D**). The 35S CaMV:GUS construct was used as a positive control. Transfection with the vector lacking a promoter (Empty) was used as a negative control. Data represent the mean ± S.E.M. of three independent replicates with three technical replicates each (*** *p* < 0.001, paired *t* test).

**Figure 6 ijms-23-15718-f006:**
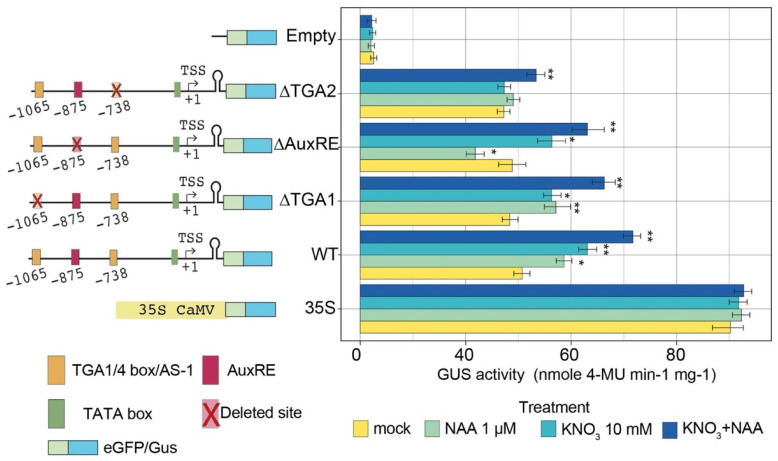
Differential contribution of TFBS within zma-*MIR528a* promoter to nitrate and auxin induction. eGFP-GUS reporter constructs under full-length zma-*MIR528a* promoter or promoters harboring deleterious mutations at either of two TGA1/4 (AS-1) or the AuxRE sites were transfected in maize protoplasts and assayed under control (mock), nitrate (10mM KNO_3_), auxin (1 µM 1-NAA), or combined treatment conditions. Deleted TFBS are shown as red-crossed boxes in each construct. Values represent GUS activity determined by fluorometry. 35S:GUS and vector lacking a promoter region (Empty) were used as positive and negative controls, respectively. Data represent mean ± S.E.M. of at least three independent experiments. One-way analysis of variance (ANOVA) and post hoc Tukey’s test were used to determine the significant difference between treatments and mock for each construct. (* *p* < 0.05; ** *p* < 0.01).

**Figure 7 ijms-23-15718-f007:**
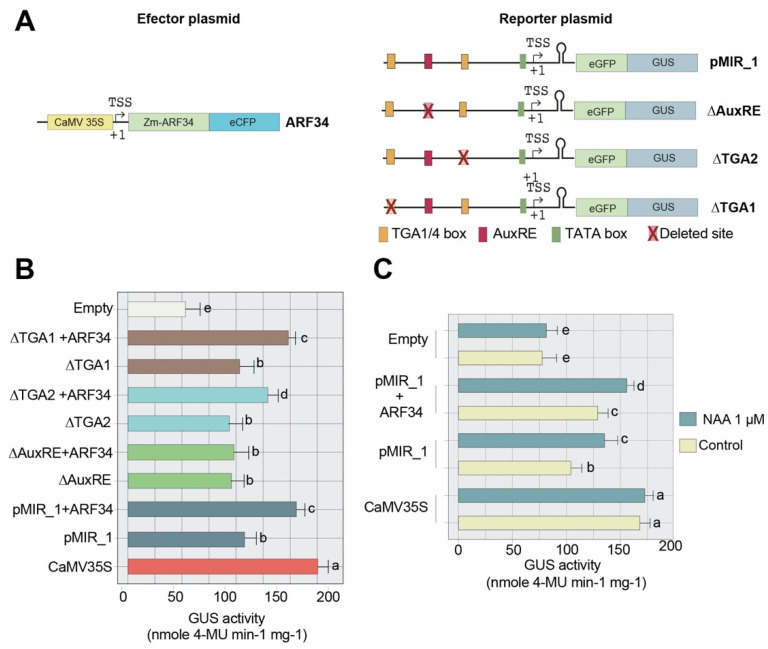
Activation of zma-*MIR528a* promoter by Zm-ARF34 (**A**) Schematic diagram of the effector and reporter constructs used in the transactivation analysis in maize protoplasts. The effector construct contained the 35S promoter fused to the ORF of ARF34 for constitutive expression. (**B**) Effect of Zm-ARF34 overexpression on the activity of the zma-*MIR528a* promoter in protoplasts. The pMIR_1, ∆AuxRE, ∆TGA2, and ∆TGA1 constructs were co-transfected with the effector plasmid in maize protoplasts incubated in control conditions. (**C**) Effect of ARF34 and auxin on the activity of the zma-*MIR528a* promoter. The pMIR_1 reporter construct was co-transfected with the expression plasmid of ARF34 in protoplasts and incubated in the absence (control) or presence of 1 µM NAA. GUS activity was normalized by absolute quantification of transfected DNA for both constructs. Values represent GUS activity determined by fluorometry. CaMV 35S:GUS and a vector lacking a promoter region (Empty) were used as positive and negative controls, respectively. Data represent mean ± S.E.M. of at least three independent experiments with three replicates each (*n* = 9). Two-way analysis of variance (ANOVA) and a post hoc multiple-comparison Tukey’s test were used to determine the significant difference between transfections and treatments. Bars that do not share at least one letter differ significantly (*p* < 0.05) from each other.

**Figure 8 ijms-23-15718-f008:**
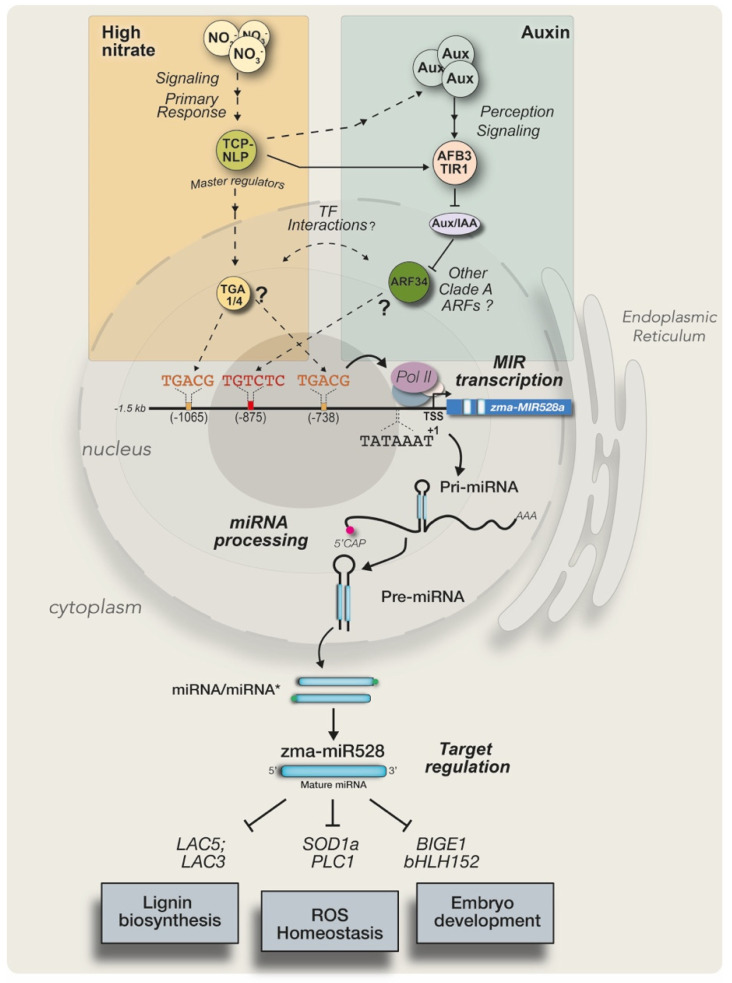
A proposed model for nitrate- and auxin-mediated transcriptional regulation over zma-*MIR528a*. Perception of high nitrate triggers primary signaling guided by master regulators to activate several TFs. TGA1, TGA4, and other TFs would recognize *cis*-regulatory elements in the promoter of zma-*MIR528a* to enhance its expression in response to the stimulus. Furthermore, elevated auxin concentration promotes the activation of AFB3 and TIR1 to polyubiquitinate the repressor Aux/IAA, setting free the clade A ARFs (transcriptional activators, such as ARF34) to interact with AuxRE at the promoter and stimulate transcription. Both pathways could overlap through TCP/NLP-mediated regulation of both auxin biosynthesis and response pathways. The nitrate signal induces auxin biosynthesis (dashed lines) as master regulators of this route promote the expression of genes involved in tryptophan metabolism to obtain IAA. These regulators and the nitrate-sensing machinery also control the nitrate–AFB3/TIR1 auxin perception to activate ARFs (solid line). Both stimuli activate *MIR528a* expression through TFs’ interaction with their binding sites. As a consequence of such coordinated regulation, miR528 level increases to down-regulate its mRNA targets.

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
