# Peer review of "Transcriptional Regulation of zma-MIR528a by Action of Nitrate and Auxin in Maize"

_ijms, 2022, doi:10.3390/ijms232415718_

Round 1

Reviewer 1 Report

Dear Authors,

The manuscript entitled “Transcriptional regulation of zma-MIR528a by action of nitrate and auxin in maize” was revised. In general, The text should be revised and formatted for the publication stage in accordance with the journal. For example, the cited references should be given in numerical order, but they are given in a mixed order. The Manuscript can be published after some corrections and minor revision.

This revisions described attached the report file

Kind regards

Author Response

All modifications are highlighted in yellow in the new manuscript version

Reviewer #1

The manuscript entitled “Transcriptional regulation of zma-MIR528a by action of nitrate and auxin in maize” was revised. In general, the text should be revised and formatted for the publication stage in accordance with the journal. For example, the cited references should be given in numerical order, but they are given in a mixed order.

R: We’re deeply sorry regarding the confusion with references. We made a mistake while formatting the manuscript. We’ve corrected the numerical order of references presented in the text to follow the journal format.

 The Manuscript can be published after some corrections and minor revision.These revisions are described below.

Title

OK

Abstract

Under this title, a little more emphasis can be placed on the purpose of the study and how it will contribute to the literature.

 R: Thanks a lot for this comment. We’ve added a sentence highlighting the study aim in the abstract section. Lines 17 and 18 in the new manuscript version.

Keywords

Since the words “Auxin”, “transcription regulation” and “zma-MIR528” were given in the title, they do not need to be given again in the keywords. Keywords are given as alternatives to the words in the title while being searched in databases. Different words will allow the article to be scanned more. These words can be omitted or replaced with miR528, AuxRE and Zea mays.

R: Thanks for your recommendation. We changed the keywords and added some others. (Line 27, page 1)

Introduction

In the first paragraph, each sentence has a lot of information, which I think is taken from one or more references. Therefore, it is not appropriate to cite this information with only two sources at the end of the paragraph.

R: We’ve taken this comment into consideration and added missing references. We also corrected the reference numbers and made the modifications in the main text. (Lines 32, 34 and 37; page 1)

Results

Line 120, ………Ogl: Oryza glaberrima, Oni: Oryza nivara, Oru: Oryza rufipogon, Osa: Oryza sativa, Pha: Panicum hallii, Pvi: Panicum virgatum,…. Species names under the same genus should be abbreviated after the first use.

R: We appreciate the reviewer’s suggestion. We’ve modified this in the new version of the manuscript in the figure legend (Figure 1, page 3).

Line 130, ….. the pri-miRNA… is this correct? Same in Line 134

 R: We’re sorry for any misunderstanding. Certainly, in both lines we referred to miR528 primary transcripts (pri-miR528a or pri-miR528b). We modified the text correspondingly (Lines 119 and 123; pages 4 and 5, respectively).

Line 184, ……… . By the other hand,….  this point belongs to the last sentence of the previous paragraph.

 R: modified in the revised manuscript version.

Discussion

OK

Material Methods

OK

Figures

OK

References

It should be carefully reviewed, the references in the text should be matched and rearranged.

R: We’ve have scrutinized and fixed the references section.

Reviewer 2 Report

In the present work, the authors aim to analyze the zma-MIR528a promoter region for identifying the conserved transcription factor binding sites. And nitrogen and auxin response elements were identified. This work provides essential information of the nitrogen and auxin-induced zma-MIR528a expression through cis-regulatory elements in its promoter, which contributes largely to our knowledge of miR528 regulome. Overall, the manuscript is clearly written. My decision is "Accept in present form".

Author Response

Reviewer #2

In the present work, the authors aim to analyze the zma-MIR528a promoter region for identifying the conserved transcription factor binding sites. And nitrogen and auxin response elements were identified. This work provides essential information of the nitrogen and auxin-induced zma-MIR528a expression through cis-regulatory elements in its promoter, which contributes largely to our knowledge of miR528 regulome. Overall, the manuscript is clearly written. My decision is "Accept in present form".

R: We appreciate the reviewer’s comment and acknowledge the professional revision of our manuscript.

Reviewer 3 Report

The authors pointed out the effect of nitrate and auxin in maize seedling development. Authors should specifically indicate the significance of their study as they have only mentioned it will be helpful in understanding regulome. I n my opinion findings are far better that what conclusion have drawn right now.

The data presented in figure 3b and 3d is not consistent. It should be re-verified.

The effect of miRNA downregulation during seed germination on its corresponding genes should also be mentioned. whether the downregulation of miRNAs increases the expression of genes it is targeting should also be presented.

Author Response

All modifications are highlighted in yellow in the new manuscript version

Reviewer # 3

The authors pointed out the effect of nitrate and auxin in maize seedling development. Authors should specifically indicate the significance of their study as they have only mentioned it will be helpful in understanding regulome. In my opinion findings are far better than what conclusion have drawn right now. 

R: We appreciate the reviewer’s comment. We added a paragraph in the discussion section regarding future perspectives and our work strengths.

The data presented in figure 3b and 3d is not consistent. It should be re-verified.

R: We’re sorry for this confusion. In Figure 3B, we qualitatively assayed miR528 precursor accumulation (pre-miR528a/b) by RT and end-point PCR, which showed a clear decrement for at least pre-miR528a and confirmed this accumulation pattern by RT-qPCR, as depicted in Figure 3C. Figure 3D refers to the accumulation profile of mature miR528 through imbibition, germination, and early seedling stages by RT-qPCR. In fact, similar accumulation profiles for mature miR528 have been reported in previous studies in maize (Li et al., 2013; Luján-Soto et al., 2021).

The effect of miRNA downregulation during seed germination on its corresponding genes should also be mentioned. Whether the downregulation of miRNAs increases the expression of genes it is targeting should also be presented.

R: We agree with the reviewer. The reduction of miR528 levels throughout imbibition and germination have a positive impact in some of its mRNA targets. This has already been reported in our previous work, where we analyzed miR528 accumulation levels during germination and assayed the correspondence between levels of some validated targets (Luján-Soto et al., 2021; Supplementary Figures, Figure S3 and Figure S4). We’ve added lines 377 to 382 discussing this.

Round 2

Reviewer 3 Report

The authors have revised the manuscript thoroughly and may be accepted for publication.